# Metagenomic data reveals type I polyketide synthase distributions across biomes

Hans W. Singh,[1] Kaitlin E. Creamer,[1] Alexander B. Chase,[1] Leesa J. Klau,[1] Sheila Podell,[1] Paul R. Jensen[1]

**ABSTRACT**  Microbial polyketide synthase (PKS) genes encode the biosynthesis of many biomedically or otherwise commercially important natural products. Despite extensive discovery efforts, metagenomic analyses suggest that only a small fraction of nature's polyketide biosynthetic potential has been realized. Much of this potential originates from type I PKSs (T1PKSs), which can be further delineated based on their domain organization and the structural features of the compounds they encode. Notably, phylogenetic relationships among ketosynthase (KS) domains provide an effective method to classify the larger and more complex T1PKS genes in which they occur. Increased access to large metagenomic data sets from diverse habitats provides opportunities to assess T1PKS biosynthetic diversity and distributions through their smaller and more tractable KS domain sequences. Here, we used the web tool NaPDoS2 to detect and classify over 35,000 type I KS domains from 137 metagenomic data sets reported from eight diverse, globally distributed biomes. We found biome-specific separation with soils enriched in KSs from modular *cis*-acetyltransferase (AT) and hybrid *cis*-AT KSs relative to other biomes and marine sediments enriched in KSs associated with polyunsaturated fatty acid and enediyne biosynthesis. We linked the phylum Actinobacteria to soil-derived enediyne and *cis*-AT KSs while marine-derived KSs associated with enediyne and monomodular PKSs were linked to phyla from which the compounds produced by these biosynthetic enzymes have not been reported. These KSs were phylogenetically distinct from those associated with experimentally characterized PKSs suggesting they may be associated with novel structures or enzyme functions. Finally, we employed our metagenome-extracted KS domains to evaluate the PCR primers commonly used to amplify type I KSs and identified modifications that could increase the KS sequence diversity recovered from amplicon libraries.

**IMPORTANCE**  Polyketides are a crucial source of medicines, agrichemicals, and other commercial products. Advances in our understanding of polyketide biosynthesis, coupled with the increased availability of metagenomic sequence data, provide new opportunities to assess polyketide biosynthetic potential across biomes. Here, we used the web tool NaPDoS2 to assess type I polyketide synthase (PKS) diversity and distributions by detecting and classifying ketosynthase (KS) domains across 137 metagenomes. We show that biomes are differentially enriched in type I KS domains, providing a roadmap for future biodiscovery strategies. Furthermore, KS phylogenies reveal biome-specific clades that do not include biochemically characterized PKSs, highlighting the biosynthetic potential of poorly explored environments. The large metagenome-derived KS data set allowed us to identify regions of commonly used type I KS PCR primers that could be modified to capture a larger extent of environmental KS diversity. These results facilitate both the search for novel polyketides and our understanding of the biogeographical distribution of PKSs across Earth's major biomes.

Address correspondence to Paul R. Jensen, pjensen@ucsd.edu.

The authors declare no conflict of interest.

See the funding table on p. 14.

**KEYWORDS**   NaPDoS2, polyketide synthase, biosynthetic diversity, natural products, specialized metabolites, metagenomes, biomes

Microorganisms are a valuable source of structurally diverse specialized metabolites, including many with clinically relevant biological activities (1, 2). Recent advances in DNA sequencing technologies and molecular genetics have fostered new discovery paradigms based on the detection of natural product biosynthetic gene clusters (BGCs) in microbial genomes (3). Instrumental to this field are online tools such as antiSMASH (4, 5) and PRISM (6) that detect and classify BGCs within query data. Additionally, the MIBiG repository (7), which lists BGCs that have been experimentally linked to compounds, and IMG-ABC (8), which details BGCs within sequenced microbial genomes, serve as important comparison points for genome mining efforts.

Polyketides represent a major source of pharmaceutically relevant specialized metabolites (9). Their biosynthesis is mediated by polyketide synthase (PKS) genes, which can be classified into types I–III, depending on their domain structure (9). Type I PKSs (T1PKSs) are composed of multidomain proteins and represent the largest source of polyketide natural products within the MIBiG repository (7). A minimal T1PKS comprises an acetyltransferase (AT) domain, which selects the appropriate building block, an acyl carrier protein (ACP) domain, to which the building block is tethered, and a ketosynthase (KS) domain, which catalyzes chain elongation between the growing polyketide and the ACP-bound extender unit (10–12). Based on the organization and function of these domains, type I PKS genes can be further delineated into two primary classes, the first of which encode enzymes that function as multimodular assembly lines (referred to here as modular *cis*-AT) where each KS domain catalyzes one round of chain elongation. *Trans*-AT PKS genes represent another version of these multimodular systems in which the AT domain occurs outside of the PKS gene (13). The second major class of T1PKSs generally has only one module (monomodular) with the KS domain functioning iteratively to catalyze more than one round of chain elongation (14).

It has long been recognized that KS phylogenies can be used to distinguish sequences associated with type I modular *cis*-AT, iterative, and *trans*-AT PKSs and thus make broader predictions about the types of PKS genes in which they occur (15–18). Type I KS phylogenies can further provide insight into the types of compounds produced (e.g., enediynes, polyunsaturated fatty acids [PUFAs], polyketide–peptide hybrids) and the functional roles of KSs in polyketide assembly (e.g., loading vs extension). The web tool NaPDoS2 (16) uses DIAMOND (19) and a well-curated reference database to detect KS domains in genomic, metagenomic, or amplicon query data. It further classifies these domains based on their top database match, which can be used to make broader predictions about PKS diversity and distributions. In this way, NaPDoS2 circumvents the need for complete PKS gene or BGC assembly, which can be particularly challenging for highly repetitive, multimodular T1PKSs, thus making it ideal for assessing biosynthetic potential within poorly assembled metagenomic data sets.

While metagenomic data have provided important new insights into natural product biosynthetic gene diversity, PCR-amplified KS domains allow for the detection of low-frequency sequences within complex assemblages. This approach has enabled the large-scale comparison of KS diversity across environmental samples (20–32) and guided the discovery of novel natural products (27). The primers used to amplify KS sequences were originally designed based on modular *cis*-AT KSs detected in the phyla Actinobacteria, Cyanobacteria, and Deltaproteobacteria (20, 21, 23). While this primer set has been modified over the years (25–27), it is unclear how well it conforms to the KS diversity now being observed in metagenomic data sets. Recent evidence that this primer set would amplify relatively few of the KS domains detected in the poorly studied phyla Acidobacteria, Verrucomicrobia, and Gemmatimonadetes suggests that modifications are warranted (33).

Assessing biosynthetic diversity using metagenomic data sets carries distinct advantages over amplicons in that complete BGCs can be captured and PCR biases

avoided. For example, work to date using metagenome-assembled genomes (MAGs) has identified previously unknown or poorly studied microbial taxa, including Acidobacteria and Candidatus Eremiobacteraeota from soils (33) and seawater (34), respectively, that are enriched in uncharacterized BGCs and thus could be targeted for natural product discovery. However, rare community members, which are an important source of natural products (33), will be poorly represented among the MAGs assembled from complex communities, with only 5.3 MAGs binned on average per metagenome in a recent analysis of 1,500 metagenomes (35). To date, metagenomes have largely been used to analyze the biosynthetic potential of individual biomes with the aim of finding new products (33, 34, 36–38). For example, direct cloning of metagenomic DNA from the human microbiome led to the discovery of new polyketide antibiotics including two that potentially play a role in microbe–microbe competition (36). In other studies, metagenomic analyses of root endophyte microbiomes led to the identification of a non-ribosomal peptide synthetase (NRPS)-PKS BGC that played a key role in disease suppression (37), while a sponge metagenome was used to link compounds to the microbes that produce them (38). Comparisons of PKS diversity across biomes are less common, although it was recently suggested, based on BGC distributions in MAGs, that specific chemistry is not limited or amplified by environment (39).

In this study, we used NaPDoS2 to detect and classify type I KS domains from eight environmental biomes. Using KS phylogenies, we detected biome-specific clades that are distinct from those associated with experimentally characterized BGCs. Additionally, we show that less than 3% on average of the KS domains in each metagenome are associated with MAGs, supporting their value as a proxy to assess biosynthetic diversity. Finally, access to environmental KS sequences provided an opportunity to evaluate the effectiveness of a widely used type I KS primer set.

## RESULTS

### Type I PKS distributions across biomes

We used NaPDoS2 to identify KS domains associated with T1PKSs in 137 shotgun metagenomes comprising 240 Gbps of data. The metagenomes represent the following eight environmental biomes: forest/agricultural soil, rhizosphere, peat soil, freshwater, seawater, freshwater sediment, marine sediment, and host associated (Table S1). In total, 35,116 KS domains were assigned to T1PKSs and an additional 409 to type I fatty acid synthases (FASs) using a minimum alignment length >200 aa. The NaPDoS2 output further delineated the non-FAS KS domains into three groups (cis-AT, trans-AT, and iterative cis-AT) and eight subgroups (hybrid cis-AT, cis-loading module, olefin synthase, PUFA, enediyne, aromatic, polycyclic tetramate macrolactam [PTM], and hybrid trans-AT) (Table S2). To validate the NaPDoS2 KS classifications, we analyzed representative sequences across the range of KS domain types by running the associated metagenomic contigs through antiSMASH 6.0 (4, 5). In each case, the KSs were associated with PKS genes that matched the NaPDoS2 classification (Fig. S1).

The majority of metagenome-extracted type I KSs (37.5%) were classified by NaPDoS2 as cis-AT with no further subgroup designation. Following that, the iterative cis-AT PUFA (20.9%) and modular cis-AT hybrid (18.9%) designations were the next most abundant (Table S2). We also analyzed the MAGs binned from each metagenome through the Joint Genome Institute (JGI) Integrated Microbial Genome (IMG) pipeline finding that, on average, only 2.7% of the type I KS domains within a given metagenome were located within MAGs (Fig. S2). This highlights the fragmented nature of the metagenomic assemblies and the utility of targeting KS sequences when assessing biosynthetic potential in complex communities.

A principal coordinates analysis (PCoA) based on a Bray–Curtis dissimilarity matrix showed a significant separation of biomes based on type I KS composition [permutational multivariate analysis of variance (ANOVA), $P < 0.001$, $R^2 = 0.499$] with PUFA, cis-AT, and hybrid cis-AT KS domains representing major drivers of biome separation between marine and non-marine samples (Fig. 1a). To further address differences in KS

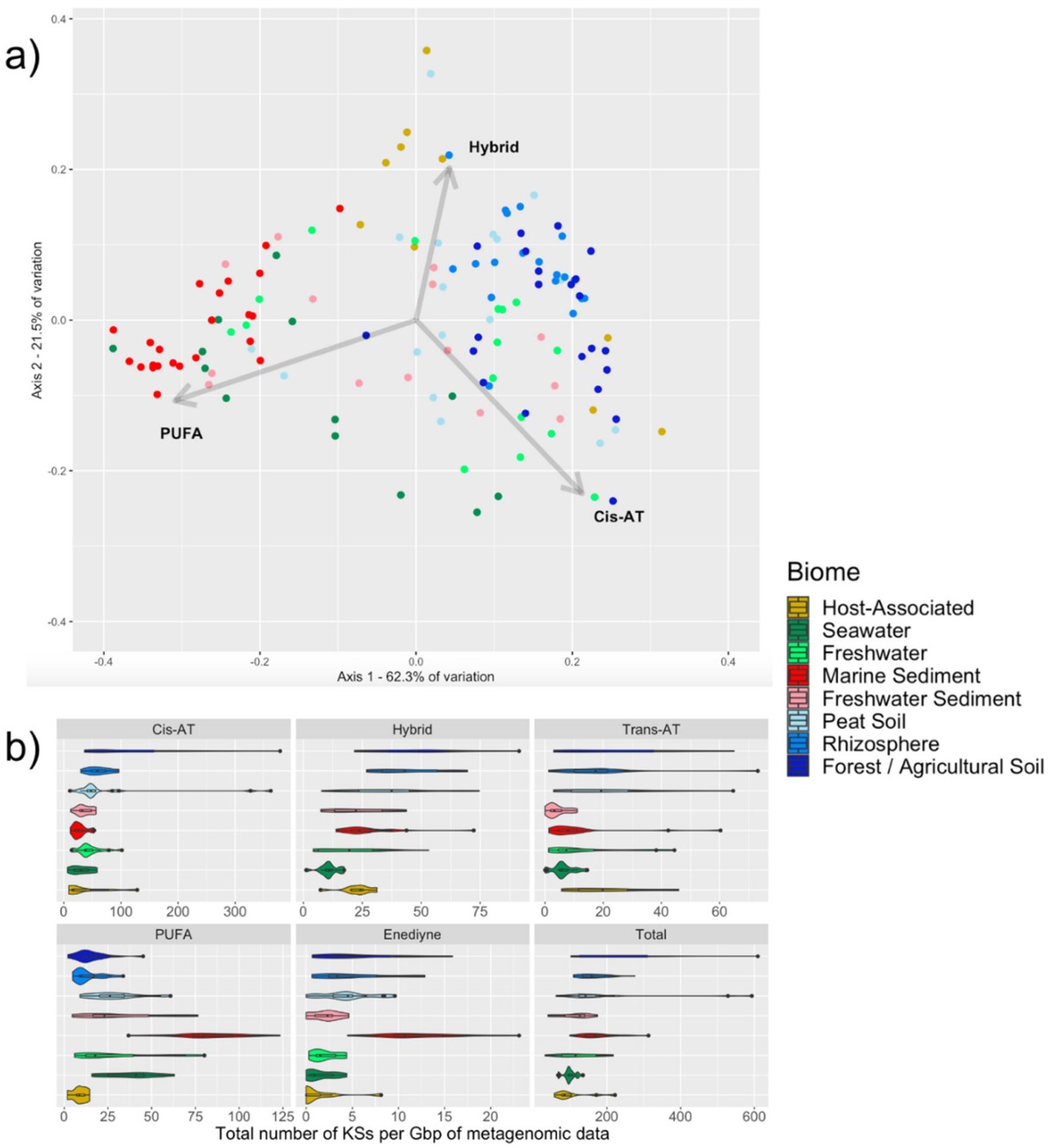

**FIG 1** Biome-specific type I KS diversity and abundance. (a) PCoA of type I KS domain distributions after transformation using a Bray–Curtis dissimilarity matrix. Each point represents a metagenomic data set (colored by biome). Arrows indicate the three KS domain types driving the most variation. (b) Violin plots showing the number of type I KS domains per Gbp of metagenomic data across the eight biomes.

composition across biomes, we determined the frequency of KSs per gigabase pair (Gbp) and found that marine sediments had significantly more PUFA and enediyne sequences (Fig. 1b) than other biomes (Tukey's honestly significant difference [HSD], $P < 0.01$). Likewise, forest/agricultural soil and rhizosphere metagenomes encoded significantly more hybrid *cis*-AT KS domains per Gbp ($P < 0.01$), and forest/agricultural soil metagenomes encoded more *cis*-AT KS domains ($P < 0.01$) than non-soil biomes (Fig. 1b).

## KS diversity across biomes

To assess KS diversity across biomes, we focused on the 7,945 full-length KS domains that could be extracted across the metagenomes as they provide a standardized framework for comparison. To assess KS richness, we clustered these sequences into operational biosynthetic units (OBUs) (40) over a range of 70%–95% amino acid sequence identity and assessed alpha diversity using Chao1 index values, a predictive measure typically applied to assess taxonomic (or operational taxonomic unit) diversity that gives more weight to rare taxa (41). Applying this approach to biosynthetic diversity, we found that soil and freshwater sediment biomes consistently carried greater OBU richness than marine sediment and seawater biomes at all clustering levels (Fig. S3). However, the only significant differences in richness were observed in forest/agricultural and peat soils, which were more diverse than non-soil biomes at the 90% and 95% clustering thresholds (Tukey's HSD, $P < 0.01$).

We next asked how the full-length KS sequences compared with those associated with experimentally characterized PKSs in the MIBiG 2.0 database. Regardless of biome, most sequences shared little similarity with the database, with only 1% overall sharing >90% amino acid sequence identity (Fig. S4). The number of KS matches dropped precipitously with increasing sequence identity across all KS types, most noticeably for hybrid and iterative PUFA KSs where 97% of the sequences had matches of <70% sequence identity. While relatively few BGCs have been experimentally characterized, this none-the-less supports the concept that considerable new polyketide diversity remains to be discovered from Earth's microbiomes. Among the few KSs that shared >90% sequence identity with the MIBiG 2.0 database, the associated BGCs represent six different biosynthetic types that account for a diverse range of natural products, including siderophores (yersiniabactin), antibiotics (e.g., rifamycin), and cyanobacterial toxins (e.g., microcystin) (Fig. S5). In our experience, KS sequence identity matches of >90% to characterized BGCs are good predictors of compound production (42, 43). The largest numbers of >90% MIBiG 2.0 sequence identity matches were observed in forest/agricultural soil, rhizosphere, host-associated, and freshwater biomes, while seawater and marine sediment had relatively few matches at this level and peat soil and freshwater sediment biomes had none (Fig. S5). This may provide insight into biomes that remain underexplored in regard to novel polyketide discovery.

We next identified the number of KS domains in OBUs shared between biomes by rarefying each biome to 580 full-length KS sequences (the lowest number in any one biome), clustering the sequences into OBUs over a range of 70%–95% amino acid sequence identity, and performing pairwise comparisons (Fig. 2). Overall, marine sediment and seawater biomes had the greatest number of KS sequences within shared OBUs at all clustering levels except 70%. Shared OBUs were also commonly identified in pairwise comparisons among forest/agricultural soil, peat soil, rhizosphere, and freshwater sediment biomes, and these always ranked among the top 10 in terms of the number of KS domains within the shared OBUs. Surprisingly, at 95% clustering, no OBUs were shared between freshwater and freshwater sediments, and only the seawater and marine sediment biomes had more than 10 KS sequences within shared OBUs. In fact, at both the 95% and 90% clustering levels, many biome combinations (64% and 46%, respectively) contained no shared OBUs, with maxima of 42 and 96 KS sequences, respectively, within the OBUs shared at these clustering levels. In contrast, at the lower clustering thresholds of 80% and 70%, all biome combinations shared at least two KS sequences, and the maximum number of shared KS sequences was 195 and 416, respectively (Fig. 2). The lack of shared OBUs between biomes at the higher sequence identity levels (90%–95%) suggests little overlap in the polyketides produced.

## Type I KS domains form five major groups

A sequence similarity network (SSN) was used to visualize relatedness among the 7,945 full-length metagenome-extracted KS domains in the context of their NaPDoS2 classification (Fig. S6). Additionally, 3,040 full-length KS domains identified in the MIBiG

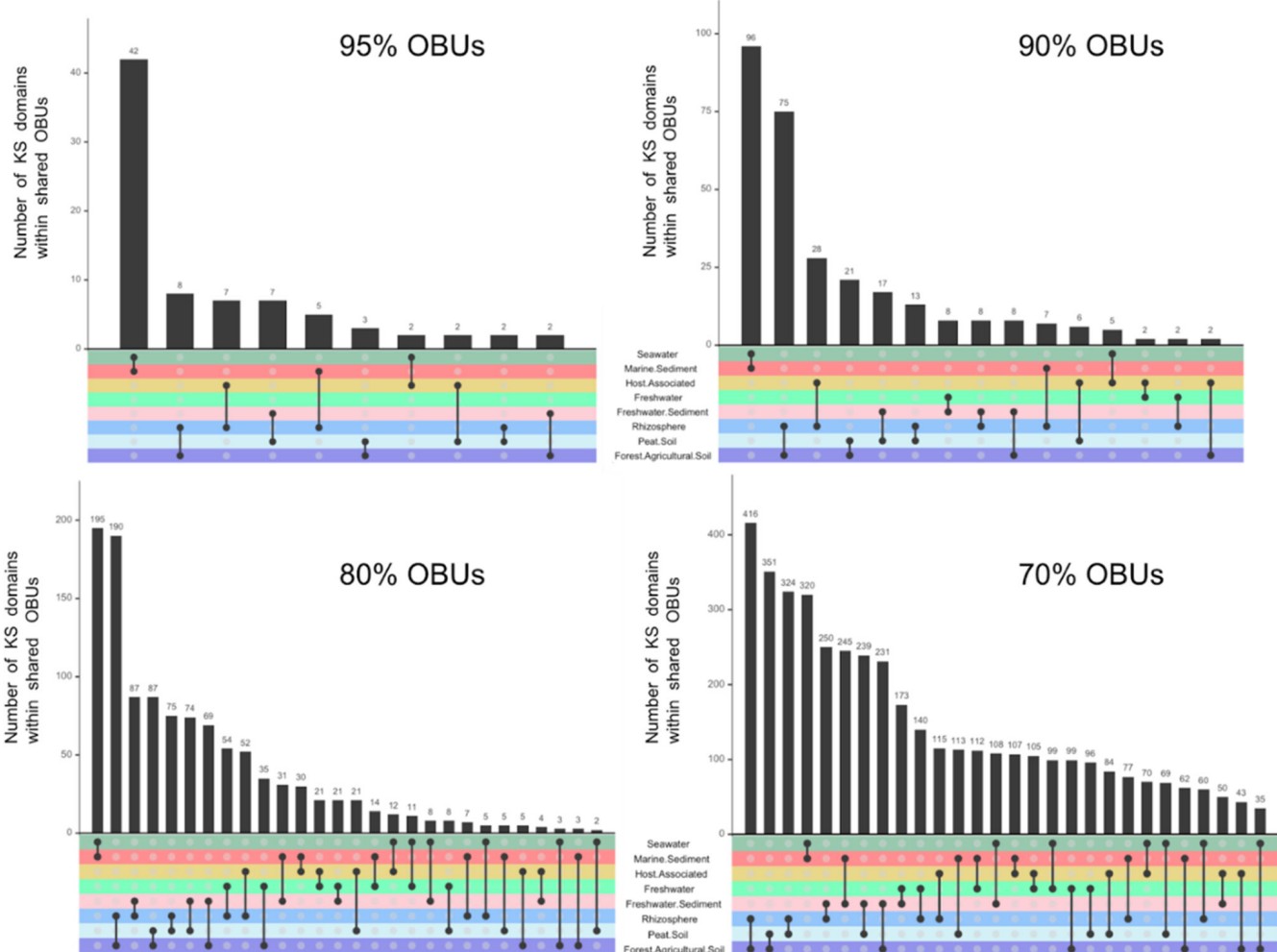

**FIG 2** KSs shared between biomes. Each biome was rarefied to 580 full-length KS sequences (the smallest number in any one biome). These sequences were clustered into operational biosynthetic units over a range (70%–95%) of amino acid sequence identities. Biome pairwise comparisons were then made and the number of KS domains within OBUs shared between biomes determined (*y*-axis). Black dots connected with a line indicate biome pairs in which shared OBUs were identified.

2.0 database were included (2,149 *cis*-AT/iterative, 725 *trans*-AT, 126 hybrid *cis*-AT, 29 PUFA, and 11 enediyne KS domains). The hybrid *cis*-AT (*n* = 1,746), *trans*-AT (*n* = 831), PUFA (*n* = 1,996), and enediyne (*n* = 210) KS domains are clearly distinguished within the SSN. The *cis*-AT/iterative cluster (*n* = 3,162) represents the fifth group and includes KSs classified by NaPDoS2 as modular (assembly line) *cis*-AT (including olefin synthase and loading module subgroups) and iteratively acting *cis*-AT (including iterative aromatic and iterative PTM subgroups). The three PUFA KS clusters correlate with the three KS domains that are usually found in PUFA PKSs (13, 24). We next generated KS phylogenies to ask if biome-specific clades could be detected within each of the five major groups identified in the SSN.

## Biome-specific and uncharacterized clades within the *cis*-AT/iterative group

Since *cis*-AT KS domains were a major driver of the separation among biomes (Fig. 1), a phylogeny was constructed using the 3,162 metagenomic and 2,149 MIBiG 2.0 sequences that comprised the *cis*-AT/iterative cluster in the SSN. This phylogeny revealed a large clade (923 sequences) in which 98.0% of the sequences mapped to soil biomes (Fig. 3a, yellow inner ring). This clade includes more than 50% of the MIBiG 2.0

reference sequences, all of which originate from multimodular, assembly-line *cis*-AT PKSs. Outside of this soil-dominant clade, the remaining 2,239 metagenome-extracted KS domains within the *cis*-AT/iterative group were more evenly spread between soil (50.5%) and non-soil (49.5%) biomes. Additionally, the phylogeny revealed a small clade of

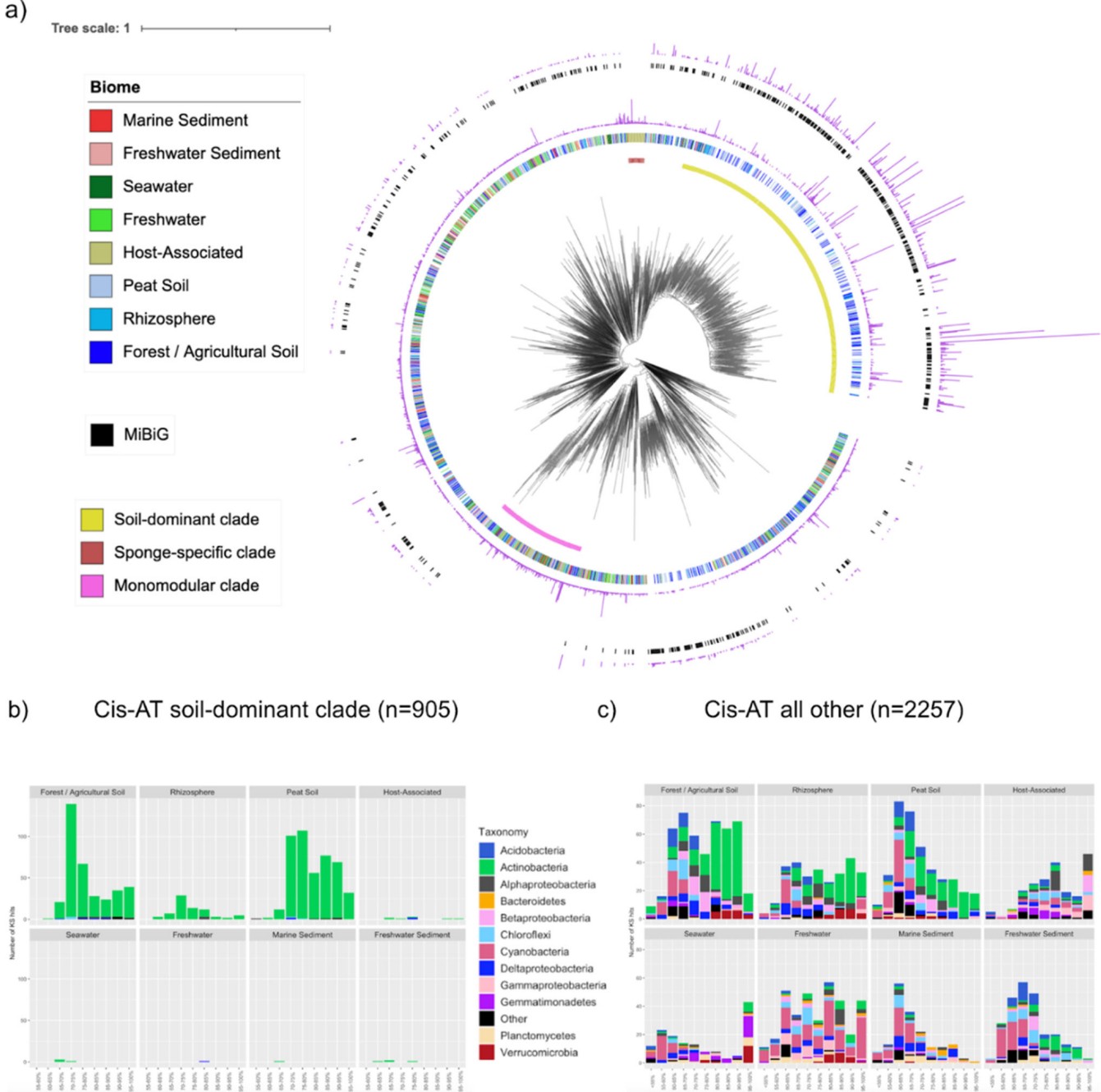

**FIG 3** Phylogeny and taxonomic distribution of KS domains from the *cis*-AT/iterative group across biomes. (a) FastME phylogeny of full-length, metagenome-extracted, *cis*-AT/iterative group KS OBUs (70% sequence identity). The innermost ring denotes the soil-dominant clade (*n* = 905, yellow), a sponge-specific clade (*n* = 89, brown), and a monomodular clade that does not include any MIBiG 2.0 sequences (*n* = 379, pink). The second ring indicates the biome from which the KS was derived, and the third ring (purple) indicates the number of metagenome-extracted KS domains in each OBU. The fourth ring (black) depicts the MIBiG 2.0 database *cis*-AT/iterative group KS domains grouped into 70% OBUs, and the fifth ring (purple) shows the number of MIBiG 2.0-extracted KS sequences in each OBU. (b) Closest Blastp taxonomic match across eight biomes for the soil-dominant clade, with the *x*-axis denoting the range of percent similarity to the closest match and the *y*-axis denoting the number of KS domains. (c) Taxonomic distributions across eight biomes for all *cis*-AT/iterative group KSs other than the soil-dominant clade with the same *x*- and *y*-axis denotations.

89 sequences that originated from sponge metagenomes (Fig. 3a, brown inner ring) and was exclusive of any MIBiG 2.0 sequences. This is consistent with previous work describing an unusual clade of T1PKSs associated with sponge symbionts (44) and KS domain sequences that predominate in sponge KS amplicon libraries (31, 32).

The taxonomic affiliations of the metagenome-extracted KS domains, assessed using the closest NCBI Blastp match, revealed that 96.2% of the *cis*-AT/iterative sequences within the soil-dominant clade could be assigned to the phylum Actinobacteria (Fig. 3b). This is not surprising given that soil-derived Actinobacteria belonging to genera, such as *Streptomyces,* have been a rich source of polyketide natural products. The sequences outside of this clade had a wider taxonomic distribution (Fig. 3c), with 23.5% assigned to Actinobacteria, 18.2% to Cyanobacteria, and 3%–9% to low abundance phyla. In contrast with the soil-dominant clade, the sponge-specific clade displayed greater taxonomic diversity, with most sequences mapping to the phyla Gemmatimonadetes (24.7%) and Alphaproteobacteria (20.2%).

The KS phylogeny of the *cis*-AT/iterative group also revealed a large clade (379 sequences) that did not group with any MIBiG 2.0 sequences (Fig. 3a, pink inner ring), indicating a lack of functional characterization. The sequences in this clade were classified by NaPDoS2 as iterative PTMs, which have been reported from Actinobacteria and Proteobacteria (6) and produce macrolactams with fused carbocyclic systems that possess wide-ranging biological activities (45) making them of interest for natural product discovery efforts. PTM biosynthesis proceeds by an unusual hybrid iterative PKS-NRPS system that contains a single module (monomodular) with the unique hybrid domain architecture of KS-AT-DH-KR-ACP-C-A-PCP-TE (45). We searched the metagenomic assemblies and found two contigs of sufficient length for antiSMASH 5.0 analysis and showed that the KSs were associated with PTM-like domain architectures (Fig. S7). We also used BLAST and NCBI RefSeq to identify related KSs (>70% sequence identity) in four phyla (Proteobacteria, Verrucomicrobia, Planctomycetes, and Bacteroidetes) and determined they were all associated with monomodular PKSs that possessed PTM-like architectures (Fig. S7). A multilocus phylogeny of these BGCs showed that the metagenome-extracted sequences were most closely related to RefSeq BGCs observed in Verrucomicrobia and Proteobacteria and distinct from the MIBiG 2.0 reference PTM BGCs (Fig. S7). The diversity and lack of MIBiG 2.0 matches among these monomodular PKSs suggest that new PTMs await discovery.

## Enediyne KS diversity across biomes

Given that marine sediments had significantly more enediyne KS sequences than other biomes (Fig. 1b), we assessed their novelty in comparison with enediyne KSs from the MIBiG 2.0 database (*n* = 11) and the NCBI RefSeq genome database (*n* = 271) after clustering into 70% OBUs. Enediynes represent a rare class of natural products that contain two acetylenic groups conjugated to a double bond within either a 9- or 10-membered ring. They are highly cytotoxic, have been developed into effective cancer drugs (46), and to date have only been reported from the phylum Actinobacteria and marine ascidian extracts (47). A phylogeny generated using the 210 full-length, metagenome-extracted enediyne KS sequences revealed a soil-specific lineage (*n* = 50) that co-localized with the genome-derived (NCBI RefSeq) sequences from Actinobacteria. This lineage encompassed all 11 MIBiG 2.0 enediyne KS domains (Fig. 4). The remaining 160 metagenome-extracted enediyne KS domains were linked to a range of phyla, including Cyanobacteria, Proteobacteria, Firmicutes, Bacteroidetes, Chloroflexi, and Spirochaetes based on their associations with the NCBI RefSeq sequences, with only two sequences of Actinobacterial origin. To further assess the non-Actinobacterial enediyne KS domains detected in RefSeq, we analyzed the respective genomes using antiSMASH 5.0 and found that all of the KS domains were associated with enediyne-like T1PKSs. Notably, 40% of the metagenome-extracted enediyne KS domains in the non-Actinobacterial portion of the phylogeny originated from marine sediments, with many (*n* = 31) observed in a large, sediment-specific clade. Sequences in this clade

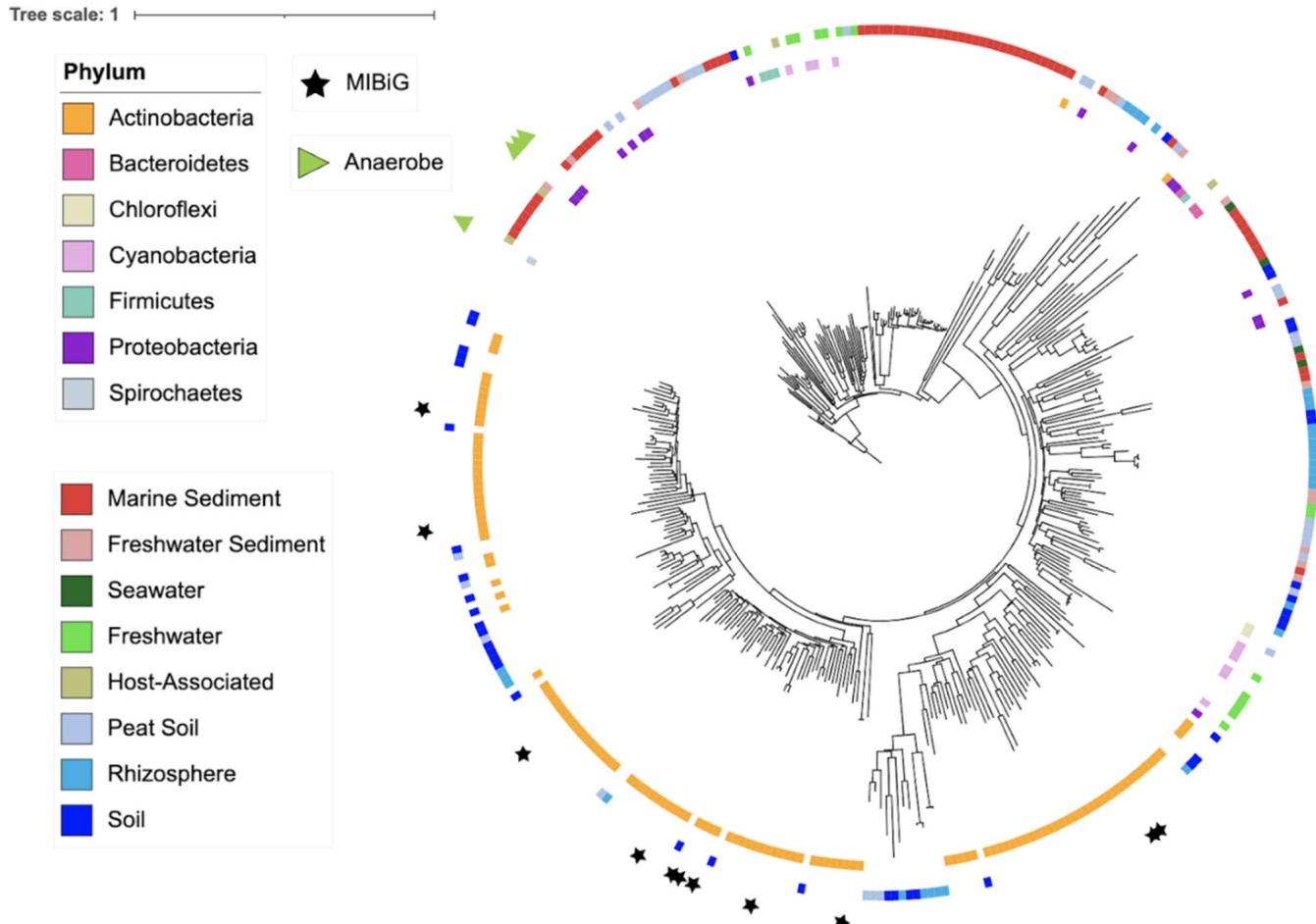

**FIG 4** Distribution of enediyne KS domains across biomes and taxa. A FastME phylogeny was built using full-length enediyne KS domains obtained from metagenomes ($n = 210$, outer ring, colored by biome), the NCBI RefSeq database (inner ring, colored by taxonomy), and the MIBiG 2.0 database (stars).

shared 85% or greater amino acid identity (NCBI BlastP) with a KS domain observed in the Deltaproteobacteria (Fig. S8), suggesting a potential new source of enediyne natural products. Interestingly, four of the RefSeq enediyne KS domains were observed in anaerobes (three from Deltaproteobacteria and one from Spirochaetes), which are also not known to produce enediyne compounds.

## Hybrid *cis*-AT, *trans*-AT, and PUFA KS domains largely lack biome specificity

We next investigated the taxonomic distributions and biome specificities of the metagenome-extracted hybrid *cis*-AT, *trans*-AT, and PUFA KS clusters identified in the SSN (Fig. S6). Hybrid *cis*-AT KS domains catalyze the condensation of an acyl group onto a PCP-tethered intermediate. *Trans*-AT KS domains occur in PKSs in which the AT domain acts in *trans* as a stand-alone AT (14). PUFA KSs occur across a wide range of bacterial phyla and contribute to the biosynthesis of linear carbon chains with multiple *cis* double bonds (24). We found little biome-specific clustering across the hybrid *cis*-AT ($n = 1,746$), *trans*-AT ($n = 831$), and PUFA KS domains ($n = 1,996$) (Fig. S9 to S11). Based on their top NCBI Blastp matches, the hybrid *cis*-AT KS domains were most often assigned to Cyanobacteria (24.7%), the *trans*-AT KS domains to Gammaproteobacteria (27.0%) and Firmicutes (21.7%), and the PUFA KS domains to Deltaproteobacteria (27.7%). From the metagenome-extracted PUFA KS domains ($n = 1,170$), over 92% fell outside of those associated with known products (Fig. S12), suggesting significant potential for the discovery and characterization of new PUFAs.

## Metagenomic KS diversity allows for the evaluation of KS PCR primers

The metagenome-extracted KS domains analyzed here are diverse in terms of their taxonomic affiliations and biome of origin. Importantly, they are not biased toward cultured strains. As such, we saw an opportunity to evaluate the commonly used type I KS primer set (KS2F/R), which has been shown to amplify *cis*-AT, *trans*-AT, and hybrid *cis*-AT KSs across diverse bacterial phyla (25–32). When comparing the amino acid specificity of the forward primer (KS2F) to our metagenome-extracted KS domains, we found it aligned best with the *cis*-AT/iterative group, with >80% of the sequences matching the amino acids targeted by the primer (Fig. S14). In contrast, only 46.0% and 40.0% of the hybrid *cis*-AT (*n* = 1,746) and *trans*-AT (*n* = 831) KS domains, respectively, matched at the third-codon position (glutamine) from the 3′ end of the KS2F primer (Fig. S14). If this primer was modified to also target histidine (H) and glutamic acid (E) at this position, these percentages would go up to 88.0% for hybrid *cis*-AT and 88.9% for *trans*-AT KSs. The KS2R reverse primer matched best with the *cis*-AT/iterative soil-dominant clade, with >90% of the sequences matching all five of the amino acids targeted by the primers. However, the other KS sequence types matched poorly with the 3′ amino acid (valine) targeted by the primer. If this 3′ position was modified to include isoleucine (I), the percentages for all KS sequence types would exceed 80%.

To date, enediyne KSs have not been reported, and PUFA KSs have rarely been reported when the KS2F/R primer set is used (29). This is consistent with our analyses, as <7% of the metagenome-extracted enediyne (*n* = 210) and PUFA (*n* = 1,996) KSs matched the amino acids targeted by the second (arginine/serine) and fourth (glutamine) positions of the forward primer (Fig. S14). We also evaluated the PUFA-specific primer set pfaA, which has been used to amplify KSs from marine sediment and seawater samples (24). The 1,170 metagenome-extracted PUFA KS domains identified by NaPDoS2 as pfaA matched well with the first five amino acids targeted by the reverse pfaA primer (glutamic acid, alanine, histidine, glycine, and threonine; Fig. S13). However, <50% of these sequences matched the amino acids targeted by the forward primer at three of the four residues closest to the 3′ end (Fig. S15).

Finally, given that PCR-generated KS amplicons are likely to be shorter than full-length sequences, we asked how this might affect the NaPDoS2 output by re-analyzing the full-length (~420 amino acids) metagenome-extracted *cis*-AT/iterative (*n* = 3,162), hybrid *cis*-AT (*n* = 1,746), and *trans*-AT (*n* = 831) KS domains after trimming them to an amplicon length typical of next-generation sequencing technologies (~138 amino acids). Overall, 94.6% of the shortened sequences yielded the same NaPDoS2 classification as their full-length counterparts (95.8% for *cis*-AT/iterative, 94.6% for hybrid *cis*-AT, and 90.5% for *trans*-AT). An SSN of the amplicon-length KS domains showed the same clustering pattern observed in the full-length KS domain SSN (Fig. S6b), further supporting the applications of NaPDoS2 for amplicon analysis.

## DISCUSSION

NaPDoS2 provides a rapid method to assess biosynthetic diversity in complex data sets by using KS and C domains to make broader predictions about PKS and NRPS genes and their small molecule products. Using the latest update to this publicly available web tool (16), which features an expanded KS classification scheme and greater capacity for large data sets, we performed an in-depth assessment of T1PKS diversity and distributions in 240 Gbp of metagenomic data representing eight environmental biomes. We observed significant differences in KS domain composition driven by PUFA KSs in marine biomes and *cis*-AT modular and hybrid *cis*-AT KSs in soil biomes (Fig. 1a). PUFAs have been suggested to aid in homeoviscous adaptation (24), which could explain why these PKSs are enriched in marine biomes. Our analyses also showed that similar biomes shared similar KS diversity, which could reflect biogeographical patterns among KS-containing microbes or environmental selection based on the functional roles of the products they encode. While a recent study of MAGs found no clear skew in relative BGC family content across Earth's microbiomes (39), we detected biome-specific variations when focusing on

diversity within type I KSs, indicating that broad surveys can obscure more subtle but potentially important environmental differences in gene content. Furthermore, we found that only a small subset (<3% on average) of the type I KS domains occurred within MAGs, highlighting the value of using KS sequences to assess biosynthetic diversity in complex communities. While the drivers of these environmental differences cannot be distinguished here, the KS diversity discerned among biomes can inform natural product discovery efforts and provide insights into the ecological roles of microbial natural products.

The majority of full-length, type I KS domains were classified as *cis*-AT with no further subclassification (Table S2), which aligns with previous genomic explorations of type I KS diversity (48). The search for biome-specific clades within the *cis*-AT/iterative group revealed a soil-dominant clade that mapped almost exclusively to Actinobacteria and grouped with *cis*-AT KS domains associated with experimentally characterized assembly-line PKSs (Fig. 3a). While experimental characterization is biased toward certain taxa, it is possible that select groups of Actinobacteria that possess large genomes are uniquely suited for assembly-line megasynthases whose polyketide products may provide a competitive advantage in soil microbiomes. Notably, these results are consistent with previous studies that have shown soil communities to be enriched in Actinobacteria compared with other biomes (49, 50). Also aligning with previous work (31, 32), our KS phylogeny revealed a sponge-specific clade that was distinct from all KS domains in the MIBiG 2.0 database (Fig. 3a), thus illustrating the potential for continued natural product discovery from sponge microbiomes. A monomodular clade that was distinct from functionally characterized sequences was also detected among the sequences classified as *cis*-AT/iterative (Fig. 3a; Fig. S5). The affiliation of these sequences with Verrucomicrobia, Planctomycetes, and Proteobacteria complements previous studies in *Streptomyces* (17) and suggests that monomodular PKSs are more widely distributed than previously recognized.

Enediyne natural products are rare and of considerable importance as anticancer drugs due to their potent cytotoxicity (46). Our analyses revealed that enediyne KSs were enriched in marine sediments relative to other biomes (Fig. 1b). A phylogeny generated from full-length KS sequences revealed affiliations with diverse phyla such as Proteobacteria, Cyanobacteria, Firmicutes, Bacteroidetes, Chloroflexi, and Spirochaetes (Fig. 4), all taxa from which this class of compounds has yet to be reported. The large marine sediment enediyne lineage is most closely related to a KS domain identified in the Deltaproteobacteria. Searching for enediyne compounds from this taxon could yield new structural diversity in a biomedically important compound class. Additionally, we report the potential for enediyne PKSs in anaerobes based on the analysis of NCBI RefSeq genomes. The phylogeny also reveals a soil-specific lineage that mapped exclusively to Actinobacteria and included all the MIBiG 2.0-derived enediyne KS domains (Fig. 4). This agrees with previous work showing that Actinobacteria account for most of the enediyne natural products described to date (46, 47).

We noticed several patterns in the taxonomic distribution KSs among bacteria. Actinobacteria were the most common taxonomic match for *cis*-AT (44%) and enediyne (28%) KS domains, whereas hybrid *cis*-AT, *trans*-AT, and PUFA domains mostly mapped to Cyanobacteria (24%), Gammaproteobacteria (28%), and Deltaproteobacteria (27%), respectively (Fig. 3; Fig. S6 to S9). In addition, 23% of the *trans*-AT KS domains mapped to Firmicutes while <4% of the other KS types mapped to this phylum. This tracks with previous reports of *trans*-AT PKSs mostly occurring in the Firmicutes and Proteobacteria phyla (51). While microbial gene databases are biased toward cultured representatives, these results suggest that bacterial phyla are differentially enriched in the types of polyketide genes they carry. Noting that 51.1% of the metagenome-extracted KS domains shared a sequence identity of less than 75% with the closest NCBI match (Fig. S13), these taxonomic assignments can be considered tentative and hint at the potential for natural product discovery from poorly studied taxa. Notably, this trend also extended when considering experimentally characterized PKSs within the MIBiG 2.0 database,

as over 92% of metagenome-extracted KS domains shared less than 75% amino acid sequence identity with the closest database match, indicating that a wealth of PKS biosynthetic diversity remains to be characterized.

Our analyses revealed that soils and freshwater sediments held greater KS richness than marine sediments and seawater, which contrasts with previous KS amplicon work that reported marine sediments to have greater KS richness than soils (29). While the optimal clustering thresholds to group KS amplicons into meaningful biosynthetic units remain unknown, the KS richness trends we observed in the metagenomes were consistent across thresholds from 70% to 95%. We also showed that when reducing full-length KS domains to next-generation amplicon lengths, 94.6% maintained the same classification (Fig. S6b), thus supporting the use of KS amplicons obtained using next-generation sequencing as proxies for full-length type I KS domains.

While metagenomes are not biased toward any specific gene, they are limited in the coverage that can be obtained when complex communities are assessed. Conversely, the targeted nature of PCR can result in a more comprehensive coverage of KS sequence diversity within a given sample, while being limited to the diversity that can be amplified by the primers. While early studies using the KS2F/R primer set revealed that soil, sponge, and sediment biomes contained significant KS diversity (25–32), a recent study found that these primers matched poorly with KSs detected in novel soil bacteria (33). Capitalizing on our metagenome-derived KS data set, we found that the KS2F/R primer set aligned best with sequences in the soil-dominant clade within the *cis*-AT/iterative group (Fig. S14). While these primers can recover some hybrid *cis*-AT and *trans*-AT KS sequences, their efficiency for these KS types could be improved with further primer design. Furthermore, the KS2F/R primer set matches poorly with both PUFA and enediyne KS domains (Fig. S14), as did a PUFA-specific primer set (24) (Fig. S15), suggesting the need for primer modifications that maximize sequence detection within these KS types.

## Conclusion

An analysis of KS domains in metagenomic data sets using NaPDoS2 revealed linkages between biosynthetic potential and environmental biomes. Through the analysis of 240 Gbps of metagenomic data, we show biome-specific differences in type I KS composition, with PUFA KSs driving the separation in marine biomes, and *cis*-AT and hybrid *cis*-AT KS domains driving the separation in soils. Furthermore, we show that similar biomes share more KS diversity than dissimilar biomes. Phylogenetic analyses of our metagenome-extracted KS domains revealed monomodular and enediyne clades that remain unexplored in terms of natural product discovery. Finally, our work revealed that the commonly used KS2F/R primer set is biased toward modular *cis*-AT KSs and is not well designed to amplify iterative *cis*-AT enediyne and PUFA KSs. This study highlights the applications of KS sequence tags to assess PKS diversity within complex metagenomic data sets.

## MATERIALS AND METHODS

### KS domain identification

One hundred and thirty-seven shotgun metagenomes representing eight biomes (agricultural/forest soil, rhizosphere, peat soil, marine sediment, freshwater sediment, seawater, host associated, and freshwater) were selected from the JGI IMG database (8) and filtered to exclude contigs <600 nucleotides using a custom script (https://github.com/spodell/NaPDoS2_website/data_management_scripts/size_limit_seqs.pl), resulting in a combined size of 240 Gbp. NaPDoS2 (16) was used to identify KS domains using a minimum match length of 200 amino acids and a minimum *E*-value of $10^{-30}$. KS domains were similarly extracted from the MIBiG 2.0 database (7), and MAGs listed on the JGI IMG database (which they assembled using MetaBAT) (8).

KS domain amino acid sequences associated with T1PKSs and FASs were identified in the NaPDoS2 output. These included the following classifications: modular *cis*-AT, *cis*-loading module, olefin synthase, iterative aromatic, iterative PTM, *trans*-AT, hybrid *trans*-AT, hybrid *cis*-AT, PUFA, enediyne, and FAS. NaPDoS2 KS classifications were verified for a randomly selected subset of metagenomic sequences across the range of KS domain types by running the associated contigs through antiSMASH 6.0 (4, 5) and comparing the output. The relative abundance of each KS domain type was calculated for each metagenome as the number of KS domains/Gbp of metagenomic data and compared using a one-way analysis of variance (ANOVA) followed by Tukey's HSD test. KS domain classifications were rarified to 100 sequences per metagenome using the average from 1,000 permutations and transformed into a Bray–Curtis dissimilarity matrix using the Vegan R program (52). This matrix was used to perform a PCoA with significant differences between biomes identified using a permutational ANOVA with the Vegan R program.

## Full-length KS domain diversity and taxonomic assignments

Full-length KS domains were filtered from the total type I metagenomic KS pool and the MIBiG 2.0 database (7). All type I KS domains within the NaPDoS2 reference database contain the start residues IAIVG and end residues GTNAH (with some degeneracy at these positions). As such, metagenomic sequences were categorized as full length if they spanned the entirety of these regions. Geneious ver. 2020.2 (53) was used for alignments. KS richness comparisons were made by randomly selecting 580 (the minimum number in any one biome) full-length KS domains from each biome using Geneious ver. 2020.2 (53) and clustering them into OBUs at 95%, 90%, 80%, and 70% amino acid sequence identity using UCLUST (54). KS richness was estimated at each sequence identity level using the Chao1 index based on the average of 10 replicate analyses and compared using a one-way ANOVA followed by Tukey's HSD test. The number of KS domains in OBUs shared between biomes was calculated using pairwise comparisons between all biome combinations. Taxonomic affiliations were assigned to full-length KS domains based on the phylum of the closest NCBI (55) Blastp ver. 2.11.0 (56) match (based on *E*-value).

## KS phylogeny

An SSN of all full-length sequences was constructed using EFI (57) with an *E*-value edge calculation of 100 and visualized using Cytoscape (58). Phylogenetic trees were constructed individually for the *trans*-AT, hybrid *cis*-AT, iterative PUFA, iterative enediyne, and *cis*-AT/iterative KS clusters identified in the SSN using FastME on the ngphylogeny.fr website (59) with default settings and visualized using iTOL (60). Due to the large number of KS domains identified in the *cis*-AT/iterative group ($n = 3,162$), they were grouped by biome and clustered into 70% OBUs using UCLUST (54). KS sequences from the MIBiG 2.0 database that were classified in the *cis*-AT/iterative group ($n = 2,149$) were similarly clustered into 70% OBUs. Centroid representatives for each OBU were used to construct FastME trees using ngphylogeny.fr (59) under default parameters and visualized using iTOL version 6 (60). Similarly, for the enediyne phylogeny, context was added to the metagenomic enediyne KSs ($n = 210$) by extracting and clustering (70% OBUs) all enediyne KS domains from the RefSeq select genomes ($n = 271$) and MIBiG 2.0 ($n = 11$) databases. No clustering was needed prior to generating the *trans*-AT ($n = 831$), hybrid *cis*-AT ($n = 1,746$), or PUFA ($n = 1,996$) phylogenies.

To visualize the genomic context of select KS domains within an uncharacterized clade in the *cis*-AT/iterative phylogeny, the relevant metagenomic contigs were analyzed using antiSMASH 5.0 (5). Since only two of what appeared to be complete BGCs were detected, related KSs that clustered into 70% OBUs with the metagenome-extracted KS domains were extracted from the NCBI RefSeq (55) protein database (release number 200) using Blastp version 2.11.0 (56). RefSeq genomes that contained these were then analyzed using antiSMASH 5.0 to identify the relevant BGCs. A multilocus phylogeny was

constructed with the metagenome-extracted BGCs, RefSeq genome-extracted BGCs, and the closest related BGCs from the MIBiG 2.0 reference database using CORASON (61).

## Evaluation of KS primers

The commonly used KS2F/KS2R (24–31) primer set is composed of the forward primer 5′-GCNATGGAYCCNCARCARMGNVT-3′ (translated to AMDPQQ(RS) (LIMV)) and the reverse primer 5′-GTNCNNGTNCCRTGNSCYTCNAC-3′ (translated to VE(AG)HGT(CWRSG)T). We aligned this primer set with the metagenome-extracted type I KS domains (amino acids), and the percent matching at each amino acid residue was calculated using Geneious ver. 2020.2 (53). The PUFA pfaA-specific primer set (24) was similarly analyzed using the metagenome-extracted KS domains that were classified as PUFA KS01 or pfaA KS domains using NaPDoS2.

## ACKNOWLEDGMENTS

This work was supported by the National Science Foundation Graduate Research Fellowship Program under Grant no. DGE-2038238 to H.W.S., DGE-1650112 to K.E.C., and the National Institutes of Health Grant no. R01GM085770 to P.R.J. Any opinions, findings, and conclusions or recommendations expressed in this material are those of the author(s) and do not necessarily reflect the views of the National Science Foundation.

We acknowledge the helpful reviewer comments and the labs that gave us permission to use their metagenomic data to assess KS distributions.

## AUTHOR AFFILIATION

[1]Center for Marine Biotechnology and Biomedicine, Scripps Institution of Oceanography, University of California San Diego, La Jolla, California, USA

## PRESENT ADDRESS

Kaitlin E. Creamer, Innovative Genomics Institute & Department of Earth and Planetary Science, University of California, Berkeley, California, USA
Alexander B. Chase, Department of Earth Sciences, Southern Methodist University, Dallas, Texas, USA
Leesa J. Klau, Department of Biotechnology and Food Science, Norwegian University of Science and Technology (NTNU), Trondheim, Norway

## AUTHOR ORCIDs

Hans W. Singh http://orcid.org/0000-0003-1453-0902
Kaitlin E. Creamer http://orcid.org/0000-0002-0666-2107
Alexander B. Chase http://orcid.org/0000-0003-1984-6279
Leesa J. Klau http://orcid.org/0000-0003-2482-3880
Sheila Podell http://orcid.org/0000-0001-7073-5190
Paul R. Jensen http://orcid.org/0000-0003-2349-1888

## FUNDING

| Funder | Grant(s) | Author(s) |
| --- | --- | --- |
| HHS | National Institutes of Health (NIH) | R01GM085770 | Paul R Jensen |
| National Science Foundation (NSF) | DGE-2038238 | Hans W. Singh |

## AUTHOR CONTRIBUTIONS

Hans W. Singh, Conceptualization, Formal analysis, Funding acquisition, Methodology, Writing – original draft, Writing – review and editing | Kaitlin E. Creamer, Formal analysis, Writing – review and editing | Alexander B. Chase, Formal analysis, Methodology, Writing

– review and editing | Leesa J. Klau, Formal analysis, Methodology, Writing – review and editing | Sheila Podell, Methodology, Writing – review and editing | Paul R. Jensen, Conceptualization, Funding acquisition, Writing – review and editing

## ADDITIONAL FILES

The following material is available online.

### Supplemental Material

**Table S1 (mSystems00012-23-s0001.pdf).** Metagenome metadata.
**Table S2; Figures S1-S15 (mSystems00012-23-s0002.pdf).** Supplemental table and figures.

### Open Peer Review

**PEER REVIEW HISTORY (review-history.pdf).** An accounting of the reviewer comments and feedback.

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
