## [Reviewer comments · mSystems]

Metagenomic Data Reveal Type I Polyketide Synthase Distributions Across Biomes

Hans Singh, Kaitlin Creamer, Alexander Chase, Leesa Klau, Sheila Podell, and Paul Jensen

Corresponding Author(s): Paul Jensen, University of California San Diego

Review Timeline:

Submission Date:	January 5, 2023
Editorial Decision:	February 9, 2023
Revision Received:	April 17, 2023
Accepted:	April 25, 2023

Editor: Gilles van Wezel

Reviewer(s): Disclosure of reviewer identity is with reference to reviewer comments included in decision letter(s). The following individuals involved in review of your submission have agreed to reveal their identity: Joleen Masschelein (Reviewer #1); Tilmann Weber (Reviewer #2)

Transaction Report:

DOI: <https://doi.org/10.1128/msystems.00012-23>

February 9, 2023

Dr. Paul R Jensen
University of California San Diego
Scripps Institution of Oceanography
9500 Gilman Drive
Mail Code 0204
La Jolla, California 92093

Re: mSystems00012-23 (Metagenomic Data Reveal Type I Polyketide Synthase Distributions Across Biomes)

Dear Paul,

Thank you for submitting your manuscript to mSystems. We have completed our review and I am pleased to inform you that, in principle, we expect to accept it for publication in mSystems. However, acceptance will not be final until you have adequately addressed the reviewer comments.

Below you will find instructions from the mSystems editorial office and comments generated during the review.

Preparing Revision Guidelines

best wishes,
Gilles van Wezel

Editor, mSystems

Journals Department
Reviewer comments:

Reviewer #1 (Comments for the Author):

This paper by Singh et al. describes the diversity and distribution of KS domains found in metagenomic data sets from diverse biomes. The authors performed extensive bioinformatics analyses to detect, classify and investigate these KS domains. Their results demonstrate that novel PKS chemistry can be discovered from metagenome-extracted KS sequences. They also reveal biome-specific separation of KS diversity and suggest that modifications can be made to the standardly used KS-specific primers to better capture this diversity. Overall, the paper is a solid piece of work that will be suitable for publication in mSystems. However, there are some issues detailed below that need to be addressed before publishing this manuscript. There are some parts of the paper that should be revised to improve clarity, for example the paragraph on the use of OBUs. In addition, the significance and implications of this paper on future studies could be elaborated more. The authors are also strongly encouraged to suggest specific modifications or improved primer sequences that could better capture KS sequence diversity from amplicon libraries.

Specific points:

- Line 31 and 34: "modular cis-AT and hybrid cis-AT KSs", "monomodular KSs": the wording here is misleading. I would rephrase this, for example as: KSs from modular and hybrid cis-AT PKSs, KSs from monomodular PKSs
- Line 38: the abstract claims that "modifications are identified that could increase the KS sequence diversity recovered from amplicon libraries". Unfortunately, however, these modifications are not clarified in the paper. It would be very valuable, both for this paper and for the larger community, if the authors could specify these modifications and suggest improved primer sequences based on their analyses.
- Line 62: The latest version is antiSMASH 6.1 with 7.0 in beta trials. There is a 2021 reference for antiSMASH 6.0. I suggest the authors insert the original 2011 reference as well.
- Line 78-82: the delineation of these three classes does not sound very logical. Trans-AT PKSs are still a multi-modular system, with just the AT domains being stand-alone. I would say the first delineation is that of iterative vs non-iterative, and then the second delineation is AT architecture.
- Line 87: please specify the abbreviation 'PUFAs'
- Line 89-90: it would be useful if the authors could provide some more information on how NaPDoS2 does this.
- Line 104-105: "Recent evidence that all 18 KS domains extracted from understudied taxa would not be amplified by this primer set suggests modifications may be warranted." This sentence needs to be changed to clarify what all 18 means here - what understudied taxa, what KS domains chosen, how large a representation is this?
- Line 113: "a poor representation..."
- Line 118: did the polyketides that were discovered from the gut microbiome also have a niche-specific role, like in the case of the root endophyte microbiomes?
- Line 121-123: "...amplified by the environment." How was this study similar or different to the one presented in this manuscript?
- Line 145: please specify the abbreviation 'PTM'
- Line 153-155: "We also analyzed the MAGs binned from each metagenome through the JGI IMG pipeline finding that, on average, only 2.7% of the type I KS domains within a given metagenome were located within MAGs". What do the authors think is the reason for this observation?
- Line 168: "KS richness and diversity across biomes" In this paragraph, it is difficult to understand the concept of the OBUs. How are the OBUs formed? When are the authors talking about individual KS sequences and when about OBUs? This is also not very clear in the supplementary information figures.
- Line 170: "To compare KS richness and diversity..." I think a more philosophical question is how do you look at KS richness and diversity here? For iterative systems there is one KS domain per pathway essentially, whereas cis-/trans-AT systems have multiple KSs per pathway. If there are five iterative KSs from five different pathways, would that count as more rich/diverse than one trans-AT system that has ten KS domains that all clade differently according to the functional group installed but assemble just one natural product?
- Line 174: 70-95% is this amino acid sequence identity?
- Line 175: it would be good if the authors could explain how this Chao1 index shows KS richness
- Figure S3: Is this figure already based on the selection of 581 KS sequences? OTU vs. OBU? Does the y-axis represent the Chao1 index? More clarification on what is compared to each other would be welcome: individual sequences or OBU? First selection of 581, followed by clustering or other way around (Discrepancy in figure vs materials and methods)? Please clarify the way this data was analyzed.
- Line 180-181: it is not clear what exactly has been done here
- Line 220-221: Please provide some additional information about this sponge-specific sup KS- clade.
- Line 237-238: KS-AT-DH-KR-C-A-TE: Please also include the ACP/PCP domain(s)
- Line 244: it would be good to include some examples of "slightly different tailoring enzymes"
- Line 247-250: Could more be made of these findings? Have any of these unknown related pathways been further studied or identified before?
- Line 419-420: "a recent study found these primers insufficient for 18 unclassified PKS pathways found in underexplored phyla" This sentence is inconsistent with what was mentioned about this in the text earlier. Reading the paper, this is the correct

interpretation but it could be clarified/rephrased.

- Line 442: the readability of the paper would improve if the methods are set in the same order as the results section
- Line 452: how was this subset of metagenomic sequences selected?
- Line 472-475: "Metagenomic sequences were considered full-length if they spanned the start residues IAIVG and end residues GTNAH (with some degeneracy at these positions) observed in all type I KS domains within the NaPDoS2 reference database with Geneious ver. 2020.2 (54) used to generate the alignments". This sentence is unclear and should be rewritten or split up in two sentences.
- Line 480: iTOL instead of ITOL
- Line 511: "...was calculated..."
- Figure 1: Where are enediyne and trans-AT KSs found in panel A?
- Line 721: "...was built..."

Reviewer #2 (Comments for the Author):

In their manuscript, Singh et al describe a comprehensive metagenome mining study for PKS encoding sequences in 137 metagenomic datasets. Instead of searching for complete PKS modules (as done by most other studies in this field), the authors used their recently updated software NaPDos, which identifies and classifies KS domains using a phylogenomic approach. Using their approach, they could identify more than 35000 KS domains in the 8 studied biomes. Interestingly, the authors found biome-specific differences in the compositions of KS-types (e.g., soils rich in cis-AT KS, marine sediments rich in enediyne and PUFAs) and also identified clades of KS, that are not yet represented with characterized BGCs in the MIBiG database. Finally, they assessed commonly used primer sets to PCR-mine for PKS-KS domains with the now enlarged dataset.

This is a straight forward descriptive study broadening our knowledge of the distribution of PKS in various environments. There are a few minor points that in my opinion should be considered to improve the manuscript:

General: The authors use a lot of abbreviations (e.g., PTM, PUFA, PCoA (why not PCA as commonly used?, OBU,...) which should be defined at their first occurrence or a "Abbreviations" table/paragraph.

Line 74ff: The sentence is a bit misleading (as type I PKSs also include trans-AT PKSs that lack the AT in the module (as the authors write themselves in Line 82/83))

Line 180-193: Are there any indications, e.g., by comparing with MIBiG hits, what metabolites these "shared" KS/PKSs biosynthesise?

Line 354ff: The authors state, that in contrast to other studies their (KS-phylogenomics-based) studied in several cases could show some biome specificity. This is a very interesting observation, but it would be nice if the authors could elucidate a bit more if the differences can be explained.

Evaluation of the KS primers:

* The primer DNA sequences should be included in the manuscript or SI

* I am wondering, why the authors based this part of the analysis on the peptide sequences of the conserved motives (which are reverse-translated to degenerate Codons in the primers). Wouldn't it make more sense to do this analysis directly on basis of the DNA sequences, which all are available in their dataset. This even may allow to do in silico PCR simulations on how the primers perform...

* Line 303ff: I find it confusing to directly connect the conserved amino acids of the KS domains with the primers, which are (degenerate) DNA sequences in phrases like "...matched all amino acid residues of the primer (line 310)" or "...matched the third amino acid residue from the 3' end of the KSF primer". Or Figure S13 which mixes DNA (5'/3') directly with amino acid motifs ("5'-EENSFP-3'")

Line 476: Please include which MIBiG version was used

Figure 1: I would highly recommend to add "line" labels for panel b - the colors are very difficult to associate with especially for lower abundances (and the color legend is reverse sorted).

Figure 3: Consider defining the "Percent range" of panel b/c in legend

Figure S1: Are there any indication, why NaPDos assigns cis-AT KS to the first example in the "Forset soil California"-case - according to the figure, it lacks the AT (and thus would be trans-AT)?

Figure S3: What does the black circle represent? (the symbols are too small to see if these are the MIBiG "stars") - but as there are only 37 transAT BGCs in MIBiGv3, I'm not sure if this the case...

Reviewer 1:

This paper by Singh et al. describes the diversity and distribution of KS domains found in metagenomic data sets from diverse biomes. The authors performed extensive bioinformatics analyses to detect, classify and investigate these KS domains. Their results demonstrate that novel PKS chemistry can be discovered from metagenome-extracted KS sequences. They also reveal biome-specific separation of KS diversity and suggest that modifications can be made to the standardly used KS-specific primers to better capture this diversity. Overall, the paper is a solid piece of work that will be suitable for publication in mSystems. However, there are some issues detailed below that need to be addressed before publishing this manuscript. There are some parts of the paper that should be revised to improve clarity, for example the paragraph on the use of OBUs.

Response (page 5, first paragraph): We have revised this paragraph about OBUs and elsewhere in the text to improve clarity.

In addition, the significance and implications of this paper on future studies could be elaborated more.

Response: we have added text throughout the paper to address this comment: page 5, line 18: "While relatively few BGCs have been experimentally characterized, this none-the-less supports the concept that considerable new polyketide diversity remains to be discovered from earth's collective microbiome." Page 5, line 24: "In our experience, KS sequence identity matches of >90% to characterized BGCs are good predictors of compound production (42, 43)." Page 5, line 45: "The lack of shared OBUs between biomes at the higher sequence identity levels (90-95%) suggests little overlap in the polyketides produced." Page 6, line 34: This is not surprising given that soil-derived Actinobacteria belonging to genera such as Streptomyces have been a rich source of polyketide natural products."

The authors are also strongly encouraged to suggest specific modifications or improved primer sequences that could better capture KS sequence diversity from amplicon libraries.

Response: Specific modifications are now included as suggested.

Specific points:

1) Line 31 and 34: "modular cis-AT and hybrid cis-AT KSs", "monomodular KSs": the wording here is misleading. I would rephrase this, for example as: KSs from modular and hybrid cis-AT PKSs, KSs from monomodular PKSs

Response: change made as suggested.

2) Line 38: the abstract claims that "modifications are identified that could increase the KS sequence diversity recovered from amplicon libraries". Unfortunately, however, these modifications are not clarified in the paper. It would be very valuable, both for this paper and for

the larger community, if the authors could specify these modifications and suggest improved primers sequences based on their analyses.

Response: The text has been revised as suggested to include primer modifications that are predicted to increase the KS diversity recovered in amplicon libraries.

3) Line 62: The latest version is antiSMASH 6.1 with 7.0 in beta trials. There is a 2021 reference for antiSMASH 6.0. I suggest the authors insert the original 2011 reference as well.

Response: Reference added as suggested.

4) Line 78-82: the delineation of these three classes does not sound very logical. Trans-AT PKSs are still a multi-modular system, with just the AT domains being stand-alone. I would say the first delineation is that of iterative vs non-iterative, and then the second delineation is AT architecture.

Response (page 2, line 41): We agreed with this comment and have revised the text as follows: "Based on the organization and function of these domains, type I PKS genes can be further delineated into two primary classes, the first of which encode enzymes that function as multi-modular assembly lines (referred to here as modular cis-AT) where each KS domain catalyzes one round of chain elongation. Trans-AT PKS genes represent another version of these multi-modular systems in which the AT domain occurs outside of the PKS gene (13). The second major class of type I PKSs generally has only one module (monomodular) with the KS domain functioning iteratively to catalyze more than one round of chain elongation (14)."

5) Line 87: please specify the abbreviation 'PUFAs'

Response: Edit made as suggested.

6) Line 89-90: it would be useful if the authors could provide some more information on how NaPDoS2 does this.

Response (page 3, line 9): The text has been revised as follows to provide more information about how NaPDoS2 classifies KS sequences: "The web tool NaPDoS2 (16) uses DIAMOND (19) and a well-curated reference database to detect KS domains in genomic, metagenomic, or amplicon query data. It further classifies these domains based on their top database match, which can be used to make broader predictions about PKS diversity and distributions."

7) Line 104-105: "Recent evidence that all 18 KS domains extracted from understudied taxa would not be amplified by this primer set suggests modifications may be warranted." This sentence needs to be changed to clarify what all 18 means here - what understudied taxa, what KS domains chosen, how large a representation is this?

Response (page 3, line 24): The text has been edited as follows: "Recent evidence that this primer set would amplify relatively few of the KS domains detected in the poorly studied phyla

Acidobacteria, Verrucomicobia, and Gemmatimonadetes suggests that modifications are warranted (33)."

8) Line 113: "a poor representation..."

Response: Change made as suggested.

9) Line 118: did the polyketides that were discovered from the gut microbiome also have a niche-specific role, like in the case of the root endophyte microbiomes?

Response (page 3, line 39): Yes. The text has been revised to the following: "For example, direct cloning of metagenomic DNA from the human microbiome led to the discovery of new polyketide antibiotics including two that potentially play a role in microbe-microbe competition (36)."

10) Line 121-123: "...amplified by the environment." How was this study similar or different to the one presented in this manuscript?

Response (page 3, line 44): The text was edited to clarify that the previous study assessed BGC diversity in MAGs whereas we analyzed KS diversity in metagenomes.

11) Line 145: please specify the abbreviation 'PTM'

Response: This has been added as suggested.

12) Line 153-155: "We also analyzed the MAGs binned from each metagenome through the JGI IMG pipeline finding that, on average, only 2.7% of the type I KS domains within a given metagenome were located within MAGs". What do the authors think is the reason for this observation?

Response (page 4, line 31): This question is addressed in the revised text: "This highlights the fragmented nature of the metagenomic assemblies and the utility of targeting KS sequences when assessing biosynthetic potential in complex communities."

13) Line 168: "KS richness and diversity across biomes" In this paragraph, it is difficult to understand the concept of the OBUs. How are the OBUs formed? When are the authors talking about individual KS sequences and when about OBUs? This is also not very clear in the supplementary information figures.

Response (page 5, line 3): The main text was revised as follows to clarify the concept of OBUs: "To assess KS richness, we clustered these sequences into operational biosynthetic units (OBUs) (40) over a range of 70-95% amino acid sequence identity and assessed alpha diversity using Chao1 index values, a predictive measure typically applied to assess taxonomic (or

operational taxonomic unit) diversity that gives more weight to rare taxa (41).". The S3 figure legend was similarly changed for clarity.

14) Line 170: "To compare KS richness and diversity..." I think a more philosophical question is how do you look at KS richness and diversity here? For iterative systems there is one KS domain per pathway essentially, whereas cis-/trans-AT systems have multiple KSs per pathway. If there are five iterative KSs from five different pathways, would that count as more rich/diverse than one trans-AT system that has ten KS domains that all code differently according to the functional group installed but assemble just one natural product?

Response: The OBU clustering largely addresses the richness aspects of this comment as the KSs in modular systems are generally highly similar and will be grouped into the same OBU. Thus, one modular PKS with multiple KSs and one iterative PKS with one KS will each be represented by one OBU. However, that does not address the absolute numbers detected per habitat, as some habitats may be enriched in modular systems and others iterative. As such, we deleted the statement that 49.7% of the full-length sequences came from soils and 50.3% came from non-soil biomes.

15) Line 174: 70-95% is this amino acid sequence identity?

Response (page 5, line 4): Yes - this has been added to the text.

16) Line 175: it would be good if the authors could explain how this Chao1 index shows KS richness

Response (page 5, line 3): The text has been modified as follows and a reference added to address this comment: "To assess KS richness, we clustered these sequences into operational biosynthetic units (OBUs) (40) over a range of 70-95% amino acid sequence identity and assessed alpha diversity using Chao1 index values, a predictive measure typically applied to assess taxonomic (or operational taxonomic unit) diversity that gives more weight to rare taxa (41)."

17) Figure S3: Is this figure already based on the selection of 581 KS sequences? OTU vs. OBU? Does the y-axis represent the Chao1 index? More clarification on what is compared to each other would be welcome: individual sequences or OBU? First selection of 581, followed by clustering or other way around (Discrepancy in figure vs materials and methods)? Please clarify the way this data was analyzed.

Response: The Figure S3 legend has been modified as follows for clarity: "Figure S3 - KS richness across biomes. Bar plot showing Chao1 KS richness across eight biomes. For each biome, 580 full-length KS domains were randomly selected and clustered into Operational Biosynthetic Units (OBUs) at four thresholds ranging from 70% to 95% amino acid sequence identity. Chao1 index values provide a predictive measure typically applied to assess taxonomic

(or operational taxonomic unit) diversity that gives more weight to rare taxa. The average of 10 analyses per biome is plotted.”

18) Line 180-181: it is not clear what exactly has been done here

Response (page 5, line 31): We recognize that this was not clear and have revised the text as follows: “We next identified the number of KS domains in OBUs shared between biomes by rarefying each biome to 580 full-length KS sequences (the lowest number in any one biome), clustering the sequences into OBUs over a range of 70-95% amino acid sequence identity, and performing pairwise comparisons (Fig. 2).”

19) Line 220-221: Please provide some additional information about this sponge-specific sup KS- clade.

Response (page 6, line 28): The following edited text and an additional reference aim to address this comment: “This is consistent with previous work describing an unusual clade of type I PKSs associated with sponge symbionts (44) and KS domain sequences that predominate in sponge KS amplicon libraries (31-32).”

20) Line 237-238: KS-AT-DH-KR-C-A-TE: Please also include the ACP/PCP domain(s)

Response: Change made as suggested.

21) Line 244: it would be good to include some examples of "slightly different tailoring enzymes"

Response (page 7, line 8): This text has been revised to the following and the figure showing the different domain organizations cited: “The RefSeq monomodular BGCs were observed in four phyla (Proteobacteria, Verrucomicrobia, Planctomycetes, and Bacteroidetes) and encoded slightly different domain architectures compared to the MIBiG 2.0 PTMs characterized to date (Fig. S7).”

22) Line 247-250: Could more be made of these findings? Have any of these unknown related pathways been further studied or identified before?

Response (page 7, line 13): To our knowledge, none of pathways from this clade have been studied. We hope this observation will draw attention to this uncharacterized biosynthetic potential.

23) Line 419-420: "a recent study found these primers insufficient for 18 unclassified PKS pathways found in underexplored phyla" This sentence is inconsistent with what was mentioned about this in the text earlier. Reading the paper, this is the correct interpretation but it could be clarified/rephrased.

Response (page 10, line 44): The text has been revised as follows for clarity: “While early studies using the KS2F/R primer set revealed that soil, sponge, and sediment biomes contained

significant KS diversity (25-32), a recent study found that these primers matched poorly with KSs detected in novel soil bacteria (33)."

24) Line 442: the readability of the paper would improve if the methods are set in the same order as the results section

Response: The methods section has been reordered and divided into additional sections to better confirm with the results section.

25) Line 452: how was this subset of metagenomic sequences selected?

Response (page 11, line 40): This has been edited to indicate that the sequences were selected randomly.

26) Line 472-475: "Metagenomic sequences were considered full-length if they spanned the start residues IAIVG and end residues GTNAH (with some degeneracy at these positions) observed in all type I KS domains within the NaPDoS2 reference database with Geneious ver. 2020.2 (54) used to generate the alignments". This sentence is unclear and should be rewritten or split up in two sentences.

Response (page 12, line 6): This text now fall into a new section and reads as follows: "Full-length KS domains were filtered from the total type I metagenomic KS pool and the MIBiG 2.0 database (7). All type I KS domains within the NaPDoS2 reference database contain the start residues IAIVG and end residues GTNAH (with some degeneracy at these positions). As such, metagenomic sequences were categorized as full-length if they spanned the entirety of these regions. Geneious ver. 2020.2 (53) was used for alignments."

27) Line 480: iTOL instead of ITOL

Response: Edit made as suggested.

28) Line 511: "...was calculated..."

Response: Edit made as suggested.

29) Figure 1: Where are enediynes and trans-AT KSs found in panel A?

Response: Each dot in this plot represents the community of KSs observed in each metagenome. The dots orient based on similarities in KS composition. The three arrows (PUFA, cis-AT, and hybrid) depict the three main KS types driving biome separation.

30) Line 721: "...was built..."

Response: Edit made as suggested.

Reviewer 2

In their manuscript, Singh et al describe a comprehensive metagenome mining study for PKS encoding sequences in 137 metagenomic datasets. Instead of searching for complete PKS modules (as done by most other studies in this field), the authors used their recently updated software NaPDos, which identifies and classifies KS domains using a phylogenomic approach. Using their approach, they could identify more than 35000 KS domains in the 8 studied biomes. Interestingly, the authors found biome-specific differences in the compositions of KS-types (e.g., soils rich in cis-AT KS, marine sediments rich in enediynes and PUFAs) and also identified clades of KS, that are not yet represented with characterized BGCs in the MIBiG database. Finally, they assessed commonly used primer sets to PCR-mine for PKS-KS domains with the now enlarged dataset.

This is a straightforward descriptive study broadening our knowledge of the distribution of PKS in various environments. There are a few minor points that in my opinion should be considered to improve the manuscript:

1) The authors use a lot of abbreviations (e.g., PTM, PUFA, PCoA (why not PCA as commonly used?, OBU,...) which should be defined at their first occurrence or a "Abbreviations" table/paragraph.

Response: we have made sure that all abbreviations are defined when first used. This has been edited as suggested. We chose a PCoA analysis since the input was a Bray-Curtis dissimilarity matrix.

2) Line 74ff: The sentence is a bit misleading (as type I PKSs also include trans-AT PKSs that lack the AT in the module (as the authors write themselves in Line 82/83))

Response (page 2, line 41): This comment was also raised by reviewer 1 and addressed with the following revisions: "Based on the organization and function of these domains, type I PKS genes can be further delineated into two primary classes, the first of which encode enzymes that function as multi-modular assembly lines (referred to here as modular cis-AT) where each KS domain catalyzes one round of chain elongation. Trans-AT PKS genes represent another version of these multi-modular systems in which the AT domain occurs outside of the PKS gene (13). The second major class of type I PKSs generally has only one module (monomodular) with the KS domain functioning iteratively to catalyze more than one round of chain elongation (14)."

3) Line 180-193: Are there any indications, e.g., by comparing with MIBiG hits, what metabolites these "shared" KS/PKSs biosynthesize?

Response: We thank the reviewer for this comment as it made us realize that we had not addressed this important point. In response, we have added a new paragraph (page 5, line 13)

and 2 new supplemental figures (S4-5) describing the novelty of the KSs and what they may produce.

4) Line 354ff: The authors state, that in contrast to other studies their (KS-phylogenomics-based) studied in several cases could show some biome specificity. This is a very interesting observation, but it would be nice if the authors could elucidate a bit more if the differences can be explained.

Response: We would need a better understanding of the ecological functions of the small molecule products of these genes to answer this question. Nonetheless, on page 9, line 16 we indicate that "PUFAs have been suggested to aid in homeoviscous adaptation (24), which could explain why these PKSs are enriched in marine biomes". This was the only speculation we felt comfortable making.

5) Evaluation of the KS primers:

* The primer DNA sequences should be included in the manuscript or SI

* I am wondering, why the authors based this part of the analysis on the peptide sequences of the conserved motives (which are reverse-translated to degenerate Codons in the primers). Wouldn't it make more sense to do this analysis directly on basis of the DNA sequences, which all are available in their dataset. This even may allow to do in silico PCR simulations on how the primers perform...

* Line 303ff: I find it confusing to directly connect the conserved amino acids of the KS domains with the primers, which are (degenerate) DNA sequences in phrases like "...matched all amino acid residues of the primer (line 310)" or "...matched the third amino acid residue from the 3' end of the KSF primer". Or Figure S13 which mixes DNA (5'/3') directly with amino acid motifs ("5'-EENSFP-3'")

Response: The primer DNA sequences have been added as suggested (page 12, line 2). We agree it would make sense to perform the analyses with DNA sequences, however the NaPDoS2 output was in amino acid format so that's what we had to work with. We also agree that the language was confusing and have edited this section to make clear that we were comparing the amino acid specificity of the primers to the amino acids in the KS sequences identified from the metagenomes.

6) Line 476: Please include which MIBiG version was used

Response: This information has been added throughout the text.

7) Figure 1: I would highly recommend to add "line" labels for panel b - the colors are very difficult to associate with especially for lower abundances (and the color legend is reverse sorted).

Response: We have re-ordered the legend so that it now matches the order presented in panel B. The legend corresponds to both panels A and B. We considered adding line numbers to

panel B and the legend, but since numbers can't be added to panel A, we felt it might create confusion when interpreting the legend. We also increased the size of the figure to help make the colors more visible.

8) Figure 3: Consider defining the "Percent range" of panel b/c in legend

Response: We have changed the X-axis legend to "percent similarity" and revised the legend for clarity as follows: (B) Closest Blastp taxonomic matches for the soil-dominant clade across eight biomes, with the x-axis denoting percent similarity to the closest match and the y-axis denoting the number of KS domains. (C) Closest Blastp taxonomic matches for all remaining cis-AT/iterative group KSs excluding the soil dominant clade across eight biomes with the same x and y-axis denotations."

9) Figure S1: Are there any indication, why NaPDos assigns cis-AT KS to the first example in the "Forest soil California"-case - according to the figure, it lacks the AT (and thus would be trans-AT)?

Response: We thank the reviewer for bringing this to our attention. We re-examined the "Forest soil California" example in the figure and determined the AH is in fact better classified as an AT domain. While we originally used annotation from the transATor webtool, which calls the domain an AH, analyses using both AntiSMASH 6 & 7 and the PKS/NRPS Analysis Web-site both call the domain an AT (the transATor score for AH versus AT is very close). In addition, we checked the PKS docking domains and they were supportive of the cis-AT annotation. Given this and the NaPDoS2 annotation as cis-AT for the KS, there is strong support to for changing the AH to an AT domain. The figure has been updated accordingly.

10) Figure S3: What does the black circle represent? (the symbols are too small to see if these are the MIBiG "stars") - but as there are only 37 transAT BGCs in MIBiGv3, I'm not sure if this the case...)

Response: We assume this comment is in reference to figure S10. The black stars do represent trans-AT KS derived from the MIBiG database (see legend). We recognize that it is difficult to recognize the shape of the stars at this magnification, but hopefully the color and similarity to figure S9, which is easier to resolve, make this clear. We also note that some trans-AT PKSs aren't labeled as such in the MIBiG display (e.g., those encoding bongkrekic acid and bryostatin as just a few examples) although they contain trans-AT KSs, thus further helping to explain the large number of stars in the outer circle.

April 25, 2023

Dr. Paul R Jensen
University of California San Diego
Scripps Institution of Oceanography
9500 Gilman Drive
Mail Code 0204
La Jolla, California 92093

Re: mSystems00012-23R1 (Metagenomic Data Reveal Type I Polyketide Synthase Distributions Across Biomes)

Dear Paul,

Your manuscript has been accepted, and I am forwarding it to the ASM Journals Department for publication. For your reference, ASM Journals' address is given below. Before it can be scheduled for publication, your manuscript will be checked by the mSystems production staff to make sure that all elements meet the technical requirements for publication. They will contact you if anything needs to be revised before copyediting and production can begin. Otherwise, you will be notified when your proofs are ready to be viewed.

If you would like to submit a potential Featured Image, please email a file and a short legend to msystems@asmusa.org. Please note that we can only consider images that (i) the authors created or own and (ii) have not been previously published. By submitting, you agree that the image can be used under the same terms as the published article. File requirements: square dimensions (4" x 4"), 300 dpi resolution, RGB colorspace, TIF file format.

We recognize that the video files can become quite large, and so to avoid quality loss ASM suggests sending the video file via <https://www.wetransfer.com/>. When you have a final version of the video and the still ready to share, please send it to mSystems staff at msystems@asmusa.org.

Sincerely,

Gilles van Wezel
Editor, mSystems

Journals Department
E-mail: mSystems@asmusa.org